# Aggregation Behavior, Antibacterial Activity and Biocompatibility of Catanionic Assemblies Based on Amino Acid-Derived Surfactants

**DOI:** 10.3390/ijms21238912

**Published:** 2020-11-24

**Authors:** Lourdes Pérez, Aurora Pinazo, M. C. Morán, Ramon Pons

**Affiliations:** 1Department of Surfactants and Nanobiotechnology, IQAC-CSIC, c/Jordi Girona, 18-26, 08034 Barcelona, Spain; aurora.pinazo@iqac.csic.es; 2Department of Biochemistry and Physiology-Physiology Section, Faculty of Pharmacy and Food Science- University of Barcelona, Avda. Joan XXIII, 27-31, 08028 Barcelona, Spain; mcmoranb@ub.edu

**Keywords:** amino acid-based surfactants, catanionic vesicles, SAXS, self-assembly, antibacterial activity, cytotoxicity

## Abstract

The surface activity, aggregates morphology, size and charge characteristics of binary catanionic mixtures containing a cationic amino acid-derived surfactant N(π), N(τ)-bis(methyl)-L-Histidine tetradecyl amide (DMHNHC_14_) and an anionic surfactant (the lysine-based surfactant N^α^-lauroyl-N^ε^acetyl lysine (C_12_C_3_L) or sodium myristate) were investigated for the first time. The cationic surfactant has an acid proton which shows a strong pK_a_ shift irrespective of aggregation. The resulting catanionic mixtures exhibited high surface activity and low critical aggregation concentration as compared with the pure constituents. Catanionic vesicles based on DMHNHC_14_/sodium myristate showed a monodisperse population of medium-size aggregates and good storage stability. According to Small-Angle X-Ray Scattering (SAXS), the characteristics of the bilayers did not depend strongly on the system composition for the positively charged vesicles. Negatively charged vesicles (cationic surfactant:myristate ratio below 1:2) had similar bilayer composition but tended to aggregate. The DMHNHC_14_-rich vesicles exhibited good antibacterial activity against Gram-positive bacteria and their bactericidal effectivity declined with the decrease of the cationic surfactant content in the mixtures. The hemolytic activity and cytotoxicity of these catanionic formulations against non-tumoral (3T3, HaCaT) and tumoral (HeLa, A431) cell lines also improved by increasing the ratio of cationic surfactant in the mixture. These results indicate that the biological activity of these systems is mainly governed by the cationic charge density, which can be modulated by changing the cationic/anionic surfactant ratio in the mixtures. Remarkably, the incorporation of cholesterol in those catanionic vesicles reduces their cytotoxicity and increases the safety of future biomedical applications of these systems.

## 1. Introduction

Mixtures of cationic and anionic surfactants, the so-called catanionic systems, have attracted considerable attention as potential alternatives to double-chain surfactants or phospholipids for the formation of vesicles for a wide range of applications [1,2]. Compared with conventional liposomes, catanionic vesicles have a number of advantages: (a) spontaneous or low energy formation, (b) high stability for long periods of time and (c) they can be prepared using simple and economic surfactants. Moreover, these catanionic mixtures, depending on the surfactant mixing ratio and total concentration, form different micro- or nano-scale aggregates such us wormlike micelles, vesicles, cubic mesophases or lamellar phases [3,4,5]. Nowadays, to better control the self-assembly of these binary mixtures, new functional surfactants are developed [6,7,8]. The number of positive or negative charges in the polar head seriously affects the micellization process of single chain quaternary surfactants [9,10,11]. Since this number is a structural characteristic that can be tuned in the pH-sensitive surfactants, one interesting strategy is the development of new surfactants sensitive to the pH as external stimuli. The micelle formation becomes progressively less favorable with the increase of cationic charges in the polar head. Critical Micellar Concentration (CMC) values clearly change when the surfactant positive charges go from 1 to 2 positive charges, however they are almost unnoticeable for 3 positive charges [9]. It has been reported that the development of pH-sensitive catanionic mixtures shows great potential for formulating drug delivery systems in which drug release is triggered by a pH change [12]. According to Lin et al., pH-sensitive vesicles can be prepared with derivatives of n-decyl phosphoric acid (DPA). The number of anionic charges in the DPA is 1 or 2 depending on the media pH. This feature allows to transform the surfactant pair in the catanionic mixture from 1/2 to 1/1 and, consequently, different self-assembled structures like micelles, vesicles and lamellar aggregates can be derived by adjusting the solution pH [13].

Catanionic mixtures can be potentially used in different industrial and pharmaceutical applications, hence combinations of non-toxic surfactants are considered as starting material to develop biocompatible and environmentally friendly mixtures. In this regard, the use of amino acid-based surfactants is a promising approach. In this work, we have used two amino acid-based surfactants (a cationic histidine derivative and an anionic lysine-based surfactant) and the anionic sodium myristate to prepare two families of biocompatible catanionic mixtures (Figure 1, Table 1).

Previous studies carried out in our group indicate that histidine is an excellent raw material to prepare cationic single-chain and gemini surfactants with remarkable physico-chemical properties and biological activity [14,15]. Histidine-derived surfactants exhibited good antibacterial activity. The gemini derivatives showed excellent surface properties and the hexadecyl homologue demonstrated its efficiency as a plasmid DNA (pDNA) and small interfering RNA (siRNA) nanocarrier [16,17]. It was also observed that the DNA/surfactant aggregates formed by these gemini surfactants did not show cytotoxicity at the concentrations required for these applications.

For the single-chain family, the DMHNHC_14_ derivative exhibited the highest activity against the microorganisms tested and also good selectivity for Gram-positive bacterial cells compared to mammalian cells. This histidine derivative has two positive charges, one of them pH-dependent. The use of this cationic pH-sensitive surfactant has some interesting advantages. First, the aggregates formed can be tuned by the pH, second, it is possible to obtain stable stoichiometric vesicles (1:1 surfactants ratio) and third, we can prepare vesicles with low content of cationic surfactants (<50%) that still have positive ζ-potential values. These characteristics can help to improve the safety of positively charged vesicles.

The physico-chemical characterization as well as the potential biomedical applications of catanionic formulations have been widely reported [1,2]. Although assessing the cytotoxicity of these systems is an important and basic step to elucidate their potential applications, their toxicological behavior has rarely been investigated. The cell viability assays showed that serine-based catanionic vesicles did not decrease the viability in A549 cells for concentrations smaller than 32 μM and that negatively charged vesicles show the smallest toxicity [3]. According to the 2,5-Diphenyl-3, -(4,5-dimethyl-2-thiazolyl) tetrazolium bromide (MTT) assay, aqueous mixtures of cationic amino acid-based surfactants and sodium dodecyl sulphate are nontoxic against NIH3T3 cells [18]. Work by Liang et al. describes that the survival of Hs68 cells was mainly controlled by the charge cationic density of oppositely charged diacetyl amphiphiles [19]. So far, we can conclude that toxicity of catanionic systems is low enough to be used in biomedical applications [20,21].

The main objectives of this work are (a) the development of stable biocompatible mixtures using affordable and environmentally friendly surfactants, and (b) the evaluation of parameters influencing the antibacterial activity and in vitro toxicity of catanionic formulations. For this purpose, we prepared catanionic formulations simply by mixing one cationic histidine-based surfactant (N(π),N(τ)-bis(methyl)-L-Histidine tetradecyl amide (DMHNHC_14_, Figure 1)) with two anionic surfactants (N^α^-lauroyl-N^ε^-acetyl-Lysine, C_12_C_3_L, and sodium myristate). Mixtures with anionic and cationic surfactants were prepared for several molar ratios (HL for DMHNHC_14_ mixtures with C_12_C_3_L and HM for those with myristate, Table 1). The aggregation process of these binary systems was characterized by surface tension, fluorescence and Small-Angle X-ray Scattering measurement and the size and charge density of the catanionic aggregates were determined by dynamic light scattering. The antibacterial activity of the designed vesicles was investigated using some representative Gram-positive and Gram-negative bacteria, also one strain of candida was tested. Moreover, erythrocytes as well as 3T3, HeLa, HaCat and A34 cells were used to gain insight on the role played by the composition of catanionic vesicles in their cytotoxicity. Finally, we have also investigated the effect of cholesterol in the biocompatibility of these colloidal systems. The implementation of these tests allowed us to determine the extent to which they can be safely used. Moreover, the experiments output will help to rationalize the design of new biocompatible catanionic mixtures.

## 2. Results and Discussion

### 2.1. Characterization of the Catanionic Systems

#### 2.1.1. Surface Tension and Acid-Base Behavior

DMHNHC_14_ surface tension was measured as a function of concentration. Results are shown in Figure 2 for three different pH conditions: unbuffered solutions, acidified and basic conditions. We expect that, in acidic conditions, solutions behave as a dicationic surfactant, whereas in basic solutions, the secondary amine is neutralized but preserves a single cationic charge on the imidazolium group.

In the three conditions described, the DMHNHC_14_ surfactant reduces the surface tension of water at small concentrations. We observe that both the unbuffered and acid pH yield similar plots. The unbuffered pH plots show a clear minimum while the plot for pH = 2 is closer to what it would be expected for a pure surfactant; that is, surface tensions plots show a break point that splits them into two parts, the first one with a negative slope and a second one with an almost constant surface tension. The break point concentration is usually identified as the critical micellar concentration (CMC) [22]. However, since surface saturation can result in formation of non-micellar aggregates, the break point does not necessarily correspond to a true CMC. Therefore, we will name this break the apparent surface tension CMC or Surface Tension critical micellar concentration (ST CMC). A minimum in the surface tension-concentration curves is usually attributed to the presence of traces of species which are more hydrophobic than the surfactant under consideration [22]. Results described in Figure 2 do not fit within this framework because if the singly charged species is that with the highest hydrophobic behavior, we would expect the minimum to occur at a somewhat smaller concentration than the ST CMC for the doubly charged surfactant. Moreover, we would expect the surface tension of the doubly charged species to be close to the unbuffered case. If this is not the case, then the presence of this minimum for the unbuffered sample should have a different origin, however, we do not have an explanation for this behavior. Additionally, it is quite notable the reduction of the apparent CMC for basic pH. The break is found more than one order of magnitude below that of the acidic of unbuffered conditions. This difference seems strong at a first glance but, a look at the scarce literature devoted to dicationic surfactants seems to confirm this trend. Examples are given in Reference [10], where the ratio CMC of single trimethylammonium compared to double head trimethylammonium bromide is 20 while the same ratio with pyridinium is 14 instead. The work in Reference [9] reports that dicationic charge long-chain C_18_ gave a ratio of 8 using an extrapolated value for the single charged species (C_18_ = 2 mM). However, in Reference [11], authors describe that the ratio for mono and dialkyl quaternary ammonium surfactants was only about 2.

To check whether there was a pK_a_ shift, we titrated the surfactant at a concentration below the CMC. The titration curve is shown in Appendix A, where 10 mL of 1.5 mM surfactant was titrated with 3 mM NaOH.

Measured pK_a_ values fall within the range 5.7 to 5.9, values that differ in about 3.3 units from those described in the literature for the histidine secondary amine. Compared to other pH-dependent surfactants, we can see that this is a large pK_a_ shift. pK_a_ values were obtained at a concentration of 1.5 mM, which is below the ST CMC of DMHNHC_14_ both in water and acidic media. At half titration, the concentration of the monocationic species would be of about 0.6 mM, which is about six times the CMC of the monocationic species. However, the pK_a_ shift is already observed in the initial titration steps and a single pK_a_ fits the whole curve. This can be seen by calculating the pK_a_ from the concentration and the initial solution pH = 4 (which results in a pK_a_ of 5.4). This implies that in the present case, the pK_a_ shift is not induced by micellization but has to be attributed to intrinsic factors, such as the presence of a nearby permanent cationic charge.

With the aim of forming catanionic vehicles, we have mixed DMHNHC_14_ with sodium myristate (Table 1). The surface tension measurements of these mixtures are shown in Figure 3. We can observe a strong decrease of the ST CMC down to values of 0.1 to 0.2 mM for all the mixtures. There is a strong synergism to decrease surface tension at lower concentrations than both the pure surfactants, in line with what was found for other catanionic systems in which the formation of the neutral pair is favored [1]. For the micellar system, this synergism can be quantified using the Rubingh method, where the interaction parameter and micellar composition can be figured out by comparing the CMC of the mixture with that of the individual surfactants [23]. Our structures are most probably vesicles, although we cannot discard the existence of mixed micelles near the apparent CMC. Applying the Rubingh method to catanionic mixtures is arguable because, in catanionic systems, the neutral composition is a true compound. Values obtained with the Rubingh method for the apparent CMC, interaction parameter β, mixed micelle composition, surface tension at CMC and apparent area per molecule are shown in Table 2.

To further discuss the Rubingh method’s applicability, we consider the different views in the literature concerning the existence of a true CMC for sodium myristate. The work in Kralchevsky et al. [24] suggests that in basic media, sodium myristate forms micelles with a CMC of 3 mM. The same authors claim that in CO_2_ saturated solutions, myristic acid crystallites coexist with monomeric sodium myristate. This contradicts previous statements concerning natural pH sodium myristate solutions [25] (the authors do not remark on any special procedure to reduce the presence of CO_2_ in their solutions). While in basic conditions, the surface tension at CMC is about 40 mNm^−1^, in 10 mM NaCl CO_2_ saturated solution, the limiting surface tension is further reduced down to about 25 mNm^−1^, with the stabilization of surface tension occurring around 0.3 mM [24]. In our results, we will not try to go further in the characterization of sodium myristate solutions, and we will use the CMC as obtained from our experiments.

The solubility of C_12_C_3_L in water was too low to determine a CMC. The surface tension of their mixtures with DMHNHC_14_ is shown in Appendix A. There is a progressive decrease of CMC with increasing the proportion of C_12_C_3_L with a CMC of 0.09 mM for the 20:80 HL mixture. The efficiency in decreasing the surface tension of the mixtures is somewhat smaller than for the HM system (see Appendix A for more detail).

#### 2.1.2. Fluorescence and Nuclear Magnetic Resonance (NMR)

We have also followed the aggregation process by fluorescence. The I_I_/I_III_ ratio is shown as a function of concentration for the pure surfactants and their mixtures in Figure 4.

While the change of polarity sensed by the probe for DMHNHC_14_ is relatively sharp, the change of polarity is progressive for all the mixtures as well as for myristate. According to Aguiar et al. [26], in this situation, the CMC value should be taken as the inflexion point (which corresponds to the point of half-height polarity and can be obtained from the fitting of a decreasing Boltzmann function). In our study, the value of X_0_/ΔX is 5.4 for DMHNHC_14_, but close to unity for the mixtures and myristate. In the article by Aguiar et al. [26], the minimum value reported is 3.3 for a nonionic surfactant with CMC 0.06 mM. This suggests that the micellization of the DMHNHC_14_ occurs in the usual cooperative way, that is, at CMC, most of the additional surfactant becomes part of the micellar pseudophase. The behavior of myristate could be congruent with the description of the aggregation process made by Kralchevsky et al. [24]. In this interpretation, there is no real CMC for the sodium myristate, with monomers coexisting with myristic acid and acid-soap crystals. The change of polarity would account for the interaction of pyrene with myristic acid and acid-soap particles which do not have a liquid hydrocarbon environment for the pyrene to be solubilized inside, but only adsorption on the surface of the particles would occur. The situation would be similar for the mixed systems, where no proper micellization occurs but direct bilayer forms as vesicles or bilayer fragments, with limited solubilization capabilities for the bulky pyrene. The values of CMC obtained by surface tension are bigger than those obtained by fluorescence for both pure compounds, while it is smaller for all the mixtures. We do not have an explanation for this behavior. Concerning the HL system, the curves of fluorescence are shown in Appendix A. All mixtures behave very closely, and the CMC values range from 0.07 to 0.15 mM.

It is clear now that the applicability of the Rubingh method is questionable in the context described. One of the components, sodium myristate, seems not to form micelles directly at room temperature [24] and there exists a neutral compound at a 1:2 mixing ratio (this would also apply in the case of mixing two micelle-forming components) and a second neutral compound at a 1:1 mixing ratio if the cationic compound has lost its acidic proton. Moreover, an open question is whether mixed micelles exist in the system or direct formation of vesicles from monomers occurs. However, we have applied the mathematical model knowing all those drawbacks and the results are given for both the surface tension and fluorescence results. While for the surface tension the mixed micelle composition calculated by the Rubingh method approaches the 1:1 behavior, the results of fluorescence tend to the 1:2 mixed micelle composition. It is noteworthy that in the case of fluorescence, the results of ζ-potential (negative for the poorer cationic surfactant mixture) qualitatively agree, while the surface tension does not account for the charge reversal in the system. A possible reason for this behavior would be the different pK_a_ shift that can occur at the air–water interface or at the micellar or vesicular surface. Because one of the origins of pK_a_ shift is the mirror charge at the interface, this effect is expected to be stronger the stronger the electrical permeability change at the interface [11]. Therefore, the molecules at the surface would lose the charge more easily than in the bulk and this would reflect in the formation of the 1:1 complex at the surface (sensed by the apparent CMC from surface tension) and the 1:2 complex in the bulk (sensed by the fluorescence measurements). It is noteworthy that this charge reversal is not progressive as it could be expected from the calculated changes in composition. The sharpness of charge reversal could be related to the low ability of counterion binding to vesicles with small excess of positively or negatively charged surfactants, which results in larger absolute values of ζ-potential than expected [27]. We did not calculate the interaction for the HL system for the absence of a usable CMC for the C_12_C_3_L.

Further insight in the structure of the system can be obtained from proton NMR. In Appendix A, different proportions of DMHNHC_14_: myristate increasing from bottom to top are shown. We can observe that both for pure DMHNHC_14_ and pure myristate, sharp peaks are obtained, while some smoothing is observed in the mixtures, notably for the two mixtures where myristate is majoritarian. This is a sign of the increase of the relaxation time of the molecules due to reduced mobility concomitant with the formation of large aggregates [28]. The two mixtures having the cationic surfactant histidine as the majoritarian component have larger concentrations of monomer or small micelles. Quantitatively, the sample with 40% histidine is the one with the lowest free surfactants, judging from the width and intensity of the peaks, agreeing with the minimum of apparent CMC.

#### 2.1.3. Dynamic Light Scattering (DLS) and ζ-Potential

ζ-potential and DLS on the same sample compositions were further used to characterize the mixed system. In Figure 5, both ζ-potential and mean diameter are shown together, with pictures showing the appearance of the samples. In general, monodisperse populations of middle size vesicles were obtained. The narrower and smaller size populations are obtained for the cationic surfactant majoritarian samples, while the rich myristate samples show bigger size distributions. They correspond respectively to clear solutions and slightly turbid bluish solutions, as illustrated in the insets of Figure 5.

The rich DMHNHC_14_ formulations could correspond to vesicles, the diameter of which is small enough to only very slightly disperse the visible light, while the myristate majoritarian are already very close and above the visible light wavelength. Also, the charge reversal is observed.

The results of size distribution of the HL system are shown in Appendix A. Very poly-disperse distributions were obtained, reflecting the low stability of the system. Sample 20:80 HL produced a relatively narrow distribution centered at 150 nm but, given the high viscosity of this composition, this peak has to be regarded as an artefact. Concerning ζ-potential of the HL systems, only positive values were obtained for the different mixtures. A value of +41 mV irrespective of composition was obtained, except for 20:80 HL that gave a viscous sample and no reliable ζ-potential values around zero. It could be questioned whether those mixed systems deserve the qualifier of catanionic. Although C_12_C_3_L is used in neutral form (acid form), when mixed with a cationic surfactant, we expect the counterions to be released to favor a neutral interface and increase the entropy of the system, as has been described by other authors [29].

The size distribution and ζ -potential values of the rich DMHNHC_14_ HM formulations suggested good storage stability. Accordingly, no phase separation was observed after 4 weeks. The rich myristate formulations, especially 20:80 HM, showed poor stability, perhaps due to the higher aggregates’ sizes and polydispersity index.

#### 2.1.4. Small-Angle X-Ray Scattering (SAXS)

The systems were also characterized by means of SAXS. Figure 6 shows the scattering curves together with lamellar best fits except for pure DMHNHC_14_, which was fitted to core-shell polydisperse spheres. The electronic profiles corresponding to these fits are also shown in Figure 6. We can observe that, although the fits are not perfect, they are quite accurate. The structure of the bilayers sensed by SAXS is quite constant, which agrees with the nearly constant “vesicle” composition obtained from fluorescence and the Rubingh model. The sample with highest myristate content shows a sharp peak at q = 1.6 nm^−1^ (second and third order reflexions visible over the background) and a secondary peak at q = 1.09 nm^−1^. The last peak can be fitted using the multilamellar model of correlated bilayers with a repetition of about 10 bilayers and repetition distance of 57 nm. The sharp peak could be due to the presence of myristate crystallites [24]. The rest of the curves could be fitted with non-correlated lamellae. Differences correspond with the different stability of both types of samples. The pure cationic surfactant could be fitted to a core shell model in which the hydrophobic radius is 1.84 nm and the hydrophilic corona thickness is 0.88 nm. This corresponds to an aggregation number of 64 and an area per molecule of 0.66 nm^2^, which fairly agrees with that measured by surface tension. The hydrophilic shell volume and electronic density agree with a hydration degree per polar head of about 17 water molecules.

The SAXS results of the HL system are shown in the Supporting Information (Appendix A). All of them showed rapid evolution, after a few seconds of exposure to X-rays. The general trend is to show a band or peak around q = 1.5 nm^−1^ which could be fitted to oligolamellar structures. The repeating distances would be of about 40 nm, which would imply poor hydration of the bilayer. One possible reason for this instability could be the formation of neutral islands that would have no charge to prevent flocculation. However, multi-lamellarity, induced by reduced charge, could also induce flocculation. The spectra were not further analyzed because of this instability.

### 2.2. Characterization of the Biological Properties

#### 2.2.1. Antibacterial Activity

Today, development of antimicrobial systems with novel modes of action against bacteria is required to minimize the alarming increase of antibiotic resistance. In this work, the antibacterial effectivity in terms of minimum inhibitory concentration (MIC) of the catanionic mixtures and their pure components is evaluated against different clinically relevant American Type Culture Collection (ATCC) strains, as shown in Table 3 and Table 4. Pure cationic histidine derivative, DMHNHC_14_, exhibited a broad spectrum of antibacterial activity. The effectivity of these surfactants against the Gram-positive bacteria is similar to that shown by the hexadecyl trimethylammonium bromide (HTAB), one of the components of commercial antiseptic formulations [30]. Sodium myristate and C_12_C_3_L did not show activity against any of the tested microorganisms, this behavior can be attributed to the null capacity of these compounds to disrupt the bacterial membrane, which acts as a selective barrier for several compounds, seemingly because these surfactants are anionic molecules, like the cell envelope itself. An effective interaction usually requires a proper electrostatic interaction between surfactants and the negative components of the bacterial membrane: lipopolysaccharides in the Gram-negative bacteria and lipoteichoic acid in the Gram-positive bacteria [31,32].

The antibacterial activity of the catanionic mixtures depends on both the proportion of cationic surfactant and the microorganism tested. The effectivity of the 80:20 HM for the Gram-positive microorganism was similar to the pure histidine derivative, decreasing with the content of cationic surfactant. Our results indicate that the bactericidal activity of the HM catanionic mixtures resulted in a lack of synergism with respect to the surfactant alone. If MIC values are measured as the DMHNHC_14_ concentration in the binary mixtures (MIC_H_), we observe that the effectivity for the 80:20 HM, 60:40 HM and 50:50 HM formulations is similar (Table 3). This suggests that the activity of these systems is governed by the concentration of cationic compound. For the rich anionic mixture (20:80 HM), the positive charge of the cationic surfactant is totally neutralized and the MIC_H_ values increase. The ζ-potential value of this formulation is negative and the interaction with the also negatively charged bacterial membrane is hindered. Recent studies revealed that the antibacterial efficiency of catanionic vesicles prepared with gemini arginine-based surfactants depends on the anionic/cationic ratio of the formulations [33].

As expected, the pure cationic surfactant as well as its HM mixtures showed higher antibacterial efficiency against Gram-positive bacteria. Usually, the inhibition of Gram-negative bacteria requires high quantity of cationic surfactants because the lipopolysaccharides present in their outer membrane make the diffusion of these compounds difficult. In fact, most of the HM formulations did not exhibit antibacterial activity against the Gram-negative bacteria at the maximum concentration tested in this assay [31].

The antibacterial activity of HL formulations showed different behavior (Table 4). In this case, both the MIC and the MIC_H_ were higher than those of pure DMHNHC_14_ for almost all strains tested. This behavior was observed for both Gram-positive and Gram-negative bacteria and the yeast *C. albicans* and could be associated to the different aggregates existing in these formulations, the very low solubility of the anionic compound in water and to the lower stability.

#### 2.2.2. Hemolytic Activity

To effectively apply catanionic systems in advanced biomedical applications, it is required to know the main factors on which their cytotoxicity depends. Since the HM catanionic systems showed better stability and solubility in phosphate-buffered saline (PBS medium) used to carry out the different cytotoxicity assays, toxicity studies were carried out using these formulations.

The first cells used to evaluate the cytotoxicity of these systems were erythrocytes, one of the most widely used cell membrane systems to study surfactant–membrane interactions [34]. Concentration-response curves corresponding to DMHNHC_14_ and its HM formulations determined from the hemolysis tests are shown in Figure 7. As expected, due to their hydrophobicity and their cationic charge, the DMHNHC_14_ hemolytic activity increases with the dose. Notice that the hemolytic activity of these cationic surfactants is lower than that of single-chain quaternary ammonium-based surfactants with similar hydrophobicity. Thus, histidine cationic surfactants are potential candidates to replace quaternary ammonium-based compounds [34].

The erythrocyte-disrupting ability of catanionic HM formulations is lower than that of the pure histidine derivative. This behavior could be related to the reduction of the cationic charge density of the aggregates in solution. In fact, the negatively charged 20:80 HM formulation showed very low hemolytic activity. Several studies have also reported that the cellular toxicity of cationic surfactants can be reduced by co-formulation with other surface-active molecules. Examples are given in Lozano et al. [35], where it is shown that the hemolysis of diacylglycerol arginine-based surfactants can be drastically reduced by preparing catanionic mixtures of this compound with phosphatidylglycerol, and the work by Chia et al. [36], that describes that the cellular viability of positively charged ionic liquids increased when these compounds were incorporated in phosphatidylcholine vesicles.

It has been reported that the incorporation of cholesterol into vesicular bilayers can modify vesicular membrane permeability and modifies their biological properties [37,38]. Based on the above considerations, we decided to study whether the addition of cholesterol to our catanionic mixtures can improve the mixtures’ biocompatibility. Figure 7B depicts the percentage of hemolysis against concentration for HM formulations containing cholesterol. Plots show that the incorporation of cholesterol to the catanionic vesicles substantially decreases the efficiency of these systems to lyse the erythrocytes membrane. Two different factors could explain this behavior: (1) the decrease of the cationic surface charge in the vesicles promoted by the addition of cholesterol, and (2) the similarity between bilayers containing cholesterol and the erythrocyte membranes. Results obtained in this work agree with those obtained recently for catanionic systems prepared with arginine-based gemini surfactants. In Pinazo et al. [33], it was found that the hemolytic activity of these formulations mainly depended on the proportion of cationic surfactant. Moreover, it was also observed that the incorporation of cholesterol into the catanionic vesicles resulted in a significant decrease in their hemolytic activity. The toxicity of the cationic dioctadecyl-dimethyl-ammonium bromide was also severely reduced when this surfactant was formulated with cholesterol and phosphatidylcholine [39].

#### 2.2.3. Cytotoxicity

The cytotoxic effects of these catanionic formulations were also evaluated using tumoral and non-tumoral cell lines. The selected cell lines include representative epidermal skin cells, such as fibroblasts (3T3 cell line) located in the dermal skin layer as well as keratinocytes (A431 and HaCaT cell lines) located in the epidermal skin layer. The study includes the HeLa cell line which constitutes the most common tumoral epithelial cell line used in short- and long-term toxicological in vitro studies on cytotoxicity and biocompatibility. Two different end points, MTT (Figure 8) and Neutral Red Uptake (NRU) (Figure 9), were used to assess differences in cell-induced cytotoxicity. Whereas the NRU method provides information about the interaction with the plasmatic membrane, the MTT method provides details about the modification of the metabolic activity of mitochondria inside the cells. Cytotoxicity assays were performed with concentrations ranging between 2.5 and 50 µM of the corresponding mixtures at four different ratios. The effect of lipid content on the formulations was evaluated by the inclusion of 20% of cholesterol. For comparative purposes, pure components at the same concentration range were evaluated.

Results show that the cytotoxic response depends strongly on both the concentration and the characteristics of the surfactant formulations (Figure 8 and Figure 9). As expected, cell viability decreases with increasing concentration of surfactants. In some cases, however, depending on both the cell line and the end point method, surfactant formulations induced proliferation, i.e., the resulting cell viability is higher than that of the control cells (cell without any treatment). The effect of charge density became an important parameter on the cell response. As a general trend, pure cationic surfactant showed the highest deleterious effect, while pure anionic surfactant induces poor modification of viability on cell cultures. Catanionic mixtures, in the presence or the absence of cholesterol, promoted intermediate responses whose values are a function of both the end point method and the cell line.

When the cellular responses were analyzed by the MTT method, cell viabilities ranged between 5–100% (3T3), 10–100% (HeLa), 25–140%(A431) and 30–130% (HaCaT) (Figure 8). In most cases, the highest value corresponds to pure myristate and the lowest one is associated with pure DMHNHC_14_. Cell viability values higher than 100% suggest that the pure myristate derivative as well as some discrete catanionic mixtures at the lowest concentrations induced proliferation on the two keratinocytes cell lines, the non-tumoral HaCaT and the tumoral A431. Proliferation was not found for 3T3 or HeLa cell lines.

Considering catanionic mixtures, the cytotoxicity decreases as the proportion of the cationic compound decreases. For the same catanionic mixture, the inclusion of cholesterol conferred protection against the cytotoxic effect of the formulations containing high content of the cationic surfactant, for which differences in cell viability ranged between 10% and 30%. An opposite behavior was observed in the case of myristate-rich formulations, for which the incorporation of the lipid reduced cell viability by around 20%.

The cellular response of these systems also depends on the end point method used to assess cell viability. A quick look at the results obtained by the NRU methods (Figure 9) demonstrated that the cellular viability was higher than those obtained by the MTT method (Figure 8). Cell viabilities ranged between 10–100% (3T3), 40–120% (A431), 40–140% (HaCaT) and 60–120% (HeLa). Besides, the effect of surfactant concentration and composition of both pure and catanionic mixtures, and the presence of cholesterol on the mixtures, promoted an attenuated trend, compared to that observed by the MTT method. As in the case of the MTT method, the lowest cell viability values were obtained with the pure cationic surfactant. In this case, however, catanionic mixtures containing the highest myristate content showed the highest cell viability values instead of the pure anionic surfactant. As a function of cell line type, cell viability did not depend on concentration, with values close to 100%, at the highest concentrations tested. In addition, the presence of cholesterol in the cationic formulations seems to induce more limited modifications.

The cellular responses upon incubation with the vesicular systems suggested that the cationic charge density can be considered one of the main controlling parameters on their cytotoxicity. This behavior is similar to that observed for the antibacterial activity of these systems. This charge is probably responsible for the initial binding of the vesicles to the negatively charged surface of the cell membrane by ionic interactions [40]. However, this binding takes place without impairing the cellular plasmatic membrane, considering the results derived from the NRU assay. As stated above, this test assesses the putative damage to the plasmatic membrane of the cells because of the interaction with the tested systems [41]. In addition, taking into account that the MTT assay is a measurement of cell metabolic activity within the mitochondrial compartment [42], the decrease in cell viability by this assay strongly suggests the uptake and internalization mediated by the vesicular system [43].

Concerning the effect of cholesterol-containing formulations, the obtained results highlighted the role of the lipid on the decrease of the whole cytotoxicity determined by either MTT or NRU methods. The inclusion of cholesterol molecules into the resulting vesicles induces dense packing of bilayers [37] that attenuate their cytotoxicity, especially for systems with the higher histidine proportions, for which the impairment was observed.

From the fitting of concentration-dependent viability curves, the corresponding concentration required to inhibit cell growth by 50% (IC_50_) of the different cell lines was determined for the different formulations. The determination of the half-maximum inhibitory concentration (IC_50_) values for the different formulations demonstrated that these binary systems can be considered biocompatible and could be suitable for different biomedical applications (Table 5). For the catanionic mixtures, the corresponding IC_50_ values resulted to be higher than the highest formulation concentration tested (IC_50_ > 50 µM) in almost all cases. The incorporation of cholesterol in the catanionic mixtures promoted higher IC_50_ values than those obtained in the absence of the lipid. Values lower than 50 µM were found only in three cases: 80:20 HM_COL HaCaT MTT (21.8 µM), 80:20 HM_COL HeLa MTT (44.5 µM) and 60:40 HM_COL 3T3 NRU (39.9 µM). The number of cellular responses with IC_50_ values < 50 µM increased to 10 for catanionic mixtures in the absence of cholesterol. IC_50_ values for discrete catanionic compositions were determined as a function of cell line and method. Only in the case of the pure histidine derivative were the corresponding IC_50_ values for all cell lines and end points found.

Results indicate that the interaction of catanionic aggregates with different biological membranes like bacteria, yeast, erythrocytes, tumoral and non-tumoral cells, is primarily dominated by their cationic charge density. This can be attributed to the fact that the cell membranes contain rich negatively charge components able to strongly interact with cationic vesicles [44]. Such conclusion agrees with several works which also reported that the surface charge density of cationic liposomes and catanionic vesicles is critical to their cellular uptake and cytotoxicity [19,36,45,46,47]

On the other hand, other physico-chemical features also seem to exert an effect on the biological activity of these colloidal systems. The rich DMHNHC_14_ vesicles showed lower turbidity related to lower mean diameter and better storage stability. These factors can also contribute to the enhanced cytotoxicity and antibacterial activity of these systems. Similarly, it was observed that the antibacterial efficiency and hemolytic activity of cationic formulations based on amino acid surfactants increases as the aggregate size decreases [48].

The results indicate that the cytotoxicity of catanionic vesicles can be tuned by changing the anionic/cationic surfactant ratio and by adding an appropriate additive. These data can help to rationalize the formulation of safer catanionic mixtures.

## 3. Materials and Methods

### 3.1. Materials

Deuterated methanol was purchased from Eurotop (Cambridge, UK). Mueller-Hinton Broth was purchased from Difco Laboratories (Detroit, MI, USA). Water from a Milli-Q Millipore system (Millipore, Burlington, MA, USA) was used to prepare aqueous solutions. Sodium myristate was from Sigma (San Luis, MO, USA). The N(π),N(τ)-bis(methyl)-L-Histidine tetradecyl amide (DMHNHC_14_) was synthesized in our laboratory following a three-step procedure [15]: (a) Synthesis of N(α)-Cbz-N(π),N(τ)-bis(methyl)-L-Histidine using (CH_3_)_2_SO_4_ (Sigma, San Luis, MO, USA) as a methylating agent, (b) preparation of the N(α)-Cbz-N(τ),N(π)-bis(methyl)-histidine tetradecyl amide simply heating the first intermediated with the tetradecyl amine (Sigma, San Luis, MO, USA) and (c) catalytic hydrogenation of intermediate 2 using Pd over charcoal (Sigma, San Luis, MO, USA). The detailed synthetic procedure as well as the characterization of DMHNHC_14_ High-Performance Liquid Chromatography (HPLC), Mass spectrometry (MS) and (NMR) are provide elsewhere [15]. The N^α^-lauroyl-N^ε^acetyl lysine (C_12_C_3_L) was obtained with a purity of 99% by the condensation of the dodecyl chloride (Sigma, San Luis, MO, USA) to the ε-amino group of the commercial acetyl-lysine. Its characterization is shown in the Supporting Information (Appendix A).

### 3.2. Catanionic Formulations

Catanionic formulations based on sodium myristate and DMHNHC_14_ were prepared by mixing aqueous solutions of pure surfactants in the appropriate ratio (Table 1). Since C_12_C_3_L is not soluble in water, mixtures containing this surfactant were prepared using a hydration method. First, solutions in methanol of C_12_C_3_L and DMHNHC_14_ were prepared. Then, 2 mL of mixtures with the corresponding quantity of each pure solution were prepared and the methanol was removed by vacuum. Finally, the obtained film was hydrated with 2 mL of Milli-Q H_2_O and sonicated at 50 °C for 15 min. Catanionic mixtures containing cholesterol were also prepared following the same hydration method procedure.

### 3.3. Surface Tension

Surface tension was measured at 25 °C using a homemade hanging drop tensiometer. Samples were hanged from a straight-cut Teflon tube and the size and shape were obtained from pictures as a function of time. Surface tension was obtained from fitting the drop profile to the appropriate Young-Laplace equations. See Sorrenti et al. [49] for more details. From a concentrated clear solution, successive dilutions were made to cover the adequate range of surface tension changes. The surface tension of DMHNHC_14_ was measured in three pH conditions: (a) unbuffered, using water as solvent, (b) buffered at pH = 2, using 10 mM HCl solution to prepare the concentrated sample and their dilutions and (c) buffered at pH = 12, using 10 mM HCl to prepare the concentrated solution and their dilutions.

### 3.4. Fluorescence Measurements

Fluorescence measurements were carried out using a Shimadzu RF 540 spectrofluorometer (Shimadzu, Kyoto, Japan). Pyrene (Sigma-Aldrich, San Luis, MO, USA) was utilized as a fluorescence probe at 25 °C. From a 5 mM solution of catanionic formulations prepared in a pyrene aqueous solution (10^−6^ M), different dilutions were prepared. The fluorescence emission spectra of these binary mixtures were acquired from 340 to 450 nm after excitation at 332 nm. The ratio of the first to the third vibronic peaks (I_I_/I_III_) in the emission spectra of pyrene is sensitive to the polarity of the microenvironment and the CMC can be calculated from plots of I_I_/I_III_ as a function of surfactant concentration. The obtained plots were fitted using a Boltzmann decreasing equation, as suggested by Aguiar et al. [26], for all the surfactants and mixtures we obtained, X0/Δx < 10 and the value of CMC correspond to the midpoint between the high and low polarity points.

### 3.5. NMR Measurements

The ^1^HNMR spectra were acquired with a Varian 400 MHz (Varian, Palo Alto, CA, USA) using a simple one-pulse experiment with a spectral width of 10,000 Hz and a pre-acquisition delay of 5 s. Cationic mixtures were prepared in deuterated water and the signal corresponding to the H_2_O impurity in the D_2_O was taken as reference. Deuterated methanol was used to carry out the ^1^HNMR and ^13^CNMR of the C_12_C_3_L compound for chemical characterization.

### 3.6. ζ-potential and size distribution analysis

The size and ζ-potential measurements of the catanionic aggregates were performed in a Malvern Zeta Nanosizer 2000 (Malvern Instruments Ltd., Malvern, UK) equipped with a He-Ne laser (633 nm, 4 mw). All analyses were done in triplicate at 25 °C. The size distribution was determined by the dynamic light scattering technique (DLS). The DLS experiment started 5 min after the sample solutions were placed in the DLS optical system to allow the sample to equilibrate at the selected temperature. The scattering intensity was measured at 173° from the incident beam. At least 10 runs were performed for each sample. The CONTIN algorithm as implemented in the instrumental software was used by the system to determine the size distribution.

The z-potential as well as the size of aggregates was measured after 24 h of preparation. Samples prepared with sodium myristate showed opalescence and no phase separation was observed after 4 weeks. The 20:80 HM precipitated after 3 days. Samples prepared with the lysine derivative showed viscosity instead of opalescence.

### 3.7. pK_a_ Determination

pKa was obtained from the titration of a solution of DMHNHC_14_ (1.5 mM) with sodium hydroxide (3 mM). The titration was followed using a glass electrode Orion coupled with an Orion Versastar reader (Thermo Scientific). The fitting of the curve showed that the sodium hydroxide was partially carbonated. 16% carbonation was determined. using Equation (1), which is derived making use of the charge and mass balances together with the acid constants of H_2_CO_3_, pK_1CO2_ = 6.35, pK_2CO2_ = 10.25 and water dissociation constant pK_w_ = 14.
(1)$$V\;=\;-\frac{V_{0}C_{a}\frac{10^{{\mathrm{pK}}_{a}-\mathrm{pH}}}{{1+10}^{{\mathrm{pK}}_{a}-\mathrm{pH}}}-V_{0}C_{a}{+V}_{0}\left(\frac{1}{10^{\mathrm{pH}}}-10^{{\mathrm{pH}+\mathrm{pK}}_{w}}\right)}{c_{b}\left(1-\alpha \left(1-\frac{1}{{1+10}^{{\mathrm{pH}+\mathrm{pK}}_{{1\mathrm{CO}}_{2}}}{+10}^{{2\mathrm{pH}+\mathrm{pK}}_{{2\mathrm{CO}}_{2}}}}\right)\right)+\left(\frac{1}{10^{\mathrm{pH}}}-10^{{\mathrm{pH}+\mathrm{pK}}_{w}}\right)}$$
where V_0_ is the titrated solution initial volume, L is concentration, pK_a_ is the searched acid constant, pk_w_ is the minus logarithm of the water dissociation constant, C_b_ is the titrant concentration, α is the fraction of carbonated titrant, pK_1CO2_ is the first dissociation constant of CO_2_ and pK_2CO2_ is the second dissociation constant of CO_2_.

### 3.8. Small-Angle X-Ray Scattering

SAXS was measured at the Non-Crystalline Diffraction beamline at ALBA Synchrotron NCD-SWEET, Barcelona. A sample-detector distance of 2.433 m was used with 0.10 nm wavelength coupled with a Q315r detector from ADSC. The q scale was calibrated with a silver behenate sample, and data reduction was performed with the software provided by the beamline. The intensity is shown as a function of scattering vector q expressed in nm^−1^. *q* = (4*π*/*λ*) sin(θ/2), where θ is the scattering angle and λ is the wavelength of radiation. The spectra have been fitted to lamellar models based on Gaussian description of the bilayer electronic profile for the vesicles, as described in Haba et al. [50]. We have kept the bilayer hydrophobic profile constant along all the samples to avoid overfitting. In these conditions, the estimated error in the position of the headgroups is 2% and 5% for the width of the Gaussian. The pure histidine derivative DMHNHC_14_ has been fitted to a core-shell model as described in Reference [28]. SAXS of the pure DMHNHC14 compound was measured using an in-house instrument S3-MICRO (Hecus X-ray systems GMBH, Graz, Austria) coupled to a GENIX-Fox three-dimensional (3D) X-ray source (Xenox, Grenoble, France), as described in Reference [50] and performing the same procedures as detailed therein.

### 3.9. Antibacterial Activity

Antibacterial assays were conducted in vitro using the broth microdilution method [51]. The minimum inhibitory concentration (MIC) is the smallest concentration of antibacterial agent at which bacterial growth is not observed. MIC values were determined for 4 Gram-positive bacteria (methicillin-resistant *Staphylococcus aureus* ATCC 43300 (MRSA), *Bacillus subtilis* ATCC 6633, *Kocuria rhizophila* 9341, *Staphylococcus epidermidis* ATCC 12228), 3 Gram-negative bacteria (*Klebsiella pneumoniae* ATCC 4532, *Pseudomonas aeruginosa* ATCC 27853, *Escherichia coli* ATCC 8739) and the fungal strain, *Candida albicans* ATCC 10231. Frozen stocks of bacteria were incubated on Muller Hinton agar plates for 24 h at 37 °C. From this culture, a bacterial suspension of 1 × 10^6^ colony forming units CFU/mL was prepared in MHB (pH = 7.3). Two dilutions of pure surfactants and their binary mixtures were prepared in MH broth to obtain a final concentration ranging from 4 to 454 μM in the microtiter plates.

Then, 10 μL of nutrient broth culture of each microorganism was added to achieve a final density of ca. 5 × 10^5^ CFU/mL. The cultures were incubated for 24 h at 37 °C. Every assay was repeated three times. Due to the low solubility, samples of anionic surfactants (C_12_C_3_L and sodium myristate) were prepared as follows: (a) preparation of a serial dilution of every compound in dimethylsulfoxide DMSO (From 9080 to 80 μM) and (b) addition of 250 μL of every solution to 4750 μL of Muller Hinton Broth to obtain a serial dilution in the culture media (from 454 to 4 μL). The two more concentrated dissolutions precipitated after 24 h, then, it was only possible to determine that the MIC values of these anionic surfactants are higher than 113 μL for all tested microorganisms.

### 3.10. Hemolysis assay and Cytotoxicity

#### 3.10.1. Hemolysis Determination

The erythrocytes, taken from rabbit blood samples, were washed three times in PBS (pH 7.4) and resuspended in PBS at a cell density of 8 × 10^9^ cells/mL. The hemolytic activity of the catanionic formulations used in this work was carried out using the procedure described by Pape et al. with minor modifications [52]. A series of different volumes of a concentrated solution (1000 μM for the general case and 5000 μM for the 20:80 and cholesterol-containing formulations) of the catanionic formulation (10–80 μL) were placed in polystyrene tubes containing 25 μL of the prepared erythrocyte suspension and PBS buffer was added to each tube to a total volume of 1 mL. The resulted solutions were incubated at room temperature and shaken for 10 min and the tubes were then centrifuged (5 min at 10,000 rpm). The hemolysis (%) was calculated by comparing the absorbance (540 nm) of the supernatant of the solutions with that of Milli-Q water (control 100% hemolysis) and PBS buffer solution (control, 0% hemolysis).

#### 3.10.2. Cell Cultures

The murine Swiss albino fibroblast (3T3), the squamous carcinoma (A431), the immortal human keratinocyte (HaCaT) and the human epithelial carcinoma (HeLa) cell lines were obtained from Celltec UB. Cells were grown in Dulbecco’s Modified Eagle’s Medium (DMEM) (4.5 g/L glucose) supplemented with 10% (*v*/*v*) fetal bovine serum (FBS), 1% (*v*/*v*) L-glutamine solution (200 mM) and 1% (*v*/*v*) penicillin-streptomycin solution (10,000 U/mL penicillin and 10 mg/mL streptomycin) at 37 °C and 5% CO_2_ (CO_2_ Incubator Nuaire, Plymouth, MN, USA). All the biological materials were purchased from Lonza (Verviers, Belgium). Cells were cultured in 75 cm^2^ culture flasks (TPP, Trasadingen, Switzerland) and were routinely split using trypsin-EDTA (ethylenediaminetetraacetate) solution (170,000 U/L trypsin and 0.2 g/L EDTA) (Lonza, Verviers, Belgium) when cells are approximately 80% confluent.

#### 3.10.3. Cell Viability Assays

3T3 and HaCaT cells (1 × 105 cells/ mL) and HeLa and A431 cells (5 × 104 cells/mL) were grown at the defined densities into the central 60 wells of a 96-well plate. Cells were incubated for 24 h under 5% CO_2_ at 37 °C. Then, the spent medium was removed, and cells were incubated during 24 h with the corresponding surfactant systems, previously diluted 50% (*v*/*v*) in DMEM medium supplemented with 5% FBS (100 µL).

#### 3.10.4. MTT Assay

In this assay, living cells reduce the yellow tetrazolium salt, 2,5-Diphenyl-3, -(4,5-dimethyl-2-thiazolyl) tetrazolium bromide (MTT), to insoluble purple formazan crystals. After the cells were incubated for 24 h with the corresponding systems, the medium was removed and 100 µL (5 mg/mL) of 2,5-Diphenyl-3,-(4,5-dimethyl-2-thiazolyl) tetrazolium bromide (MTT) (Sigma–Aldrich, St. Louis, MO, USA) in phosphate-buffered saline (PBS) (Lonza, Verviers, Belgium), diluted 1:10 in culture medium without phenol red and absence of FBS (Lonza, Verviers, Belgium), was added to the cells. The plates were incubated for 3 h, after which the medium was removed. Thereafter, 100 µL of dimethylsulfoxide (DMSO) (Sigma–Aldrich, St. Louis, MO, USA) was added to each well to dissolve the purple formazan crystals. Plates were then placed in a microtiter-plate shaker for 5 min at room temperature to help the total dissolution. Then, the absorbance of the resulting solutions was measured at 550 nm using a Bio-Rad 550 microplate reader (Hercules, California, USA). The effect of each treatment was calculated as the percentage of tetrazolium salt reduction by viable cells against the untreated cell control (cells with medium only).

#### 3.10.5. NRU Assay

The accumulation of the neutral red dye in the lysosomes of viable, undamaged cells constitutes the basis of the Neutral Red Uptake (NRU) assay. After the cells were incubated for 24 h with the corresponding systems, the medium was removed, and the cells were incubated for 3 h with the NRU reagent (Sigma–Aldrich, St. Louis, MO, USA) solution (50 µg/mL) dissolved in the medium without FBS and phenol red (Lonza, Verviers, Belgium). Cells were then washed with sterile phosphate-buffered saline (PBS) (Lonza, Verviers, Belgium), followed by the addition of 100 µL of a solution containing 50% ethanol absolute and 1% acetic acid in distilled water to extract the dye. Agitation, determination of the absorbance of the extracted solution and effect of each treatment in comparison with control cells were analogous of those described in the case of the MTT method.

## 4. Conclusions

This study described the development of stable catanionic vesicles in mixtures of cationic amino acid-based surfactant mixed with an anionic amino acid-based surfactant or with sodium myristate. The CMC values of these mixtures have been measured by surface tension and fluorescence. All mixtures showed a reduction of CMC. These synergistic effects were evaluated using the Rubingh method obtaining negative values for the interaction parameters and values for the composition of the aggregates. When the aggregation was determined by surface tension, the resulting composition tended to 1:1, but the composition tended to 1:2 when the calculations were based on fluorescence. Fluorescence results qualitatively agreed with ζ-potential. DLS and SAXS studies indicated that all the mixtures can be described as vesicles, most of them in unilamellar form. Systems that can be described as oligolamellar vesicles showed poor stability. The composition of the bilayer does not change very much with system composition, agreeing with the results obtained from the Rubingh model.

The biological properties of the DMHNHC_14_/sodium myristate mixtures have been determined as a function of the mixing ratio. The rich DMHNHC_14_ formulations exhibited good antibacterial activity against Gram-positive bacteria. The cytotoxicity against erythrocytes, fibroblasts (3T3 cell line), keratinocytes (A431 and HaCaT cell lines) and the well-known HeLa cell line as a model of tumoral epithelial cells, depends on the anionic/cationic surfactant ratio, the cell line and the method used to assess cell viability. As a general trend, the antibacterial activity and cytotoxicity of these systems increases by increasing the cationic charge density in the mixtures. The surfactant mixing ratio allows the tuning of the surface charge of the vesicles; therefore, the biological activity of these formulations can be easily modulated. Interestingly, the incorporation of cholesterol in catanionic vesicles can reduce their cytotoxicity and increase the safety for future applications of these systems in biomedical applications.

## Figures and Tables

**Figure 1 ijms-21-08912-f001:**
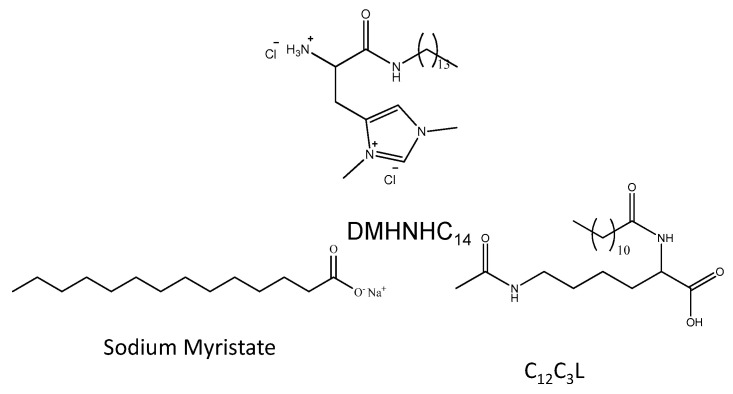
Chemical structure of the cationic (DMHNHC_14_) and anionic (C_12_C_3_L, sodium myristate) surfactants used to prepare the catanionic mixtures.

**Figure 2 ijms-21-08912-f002:**
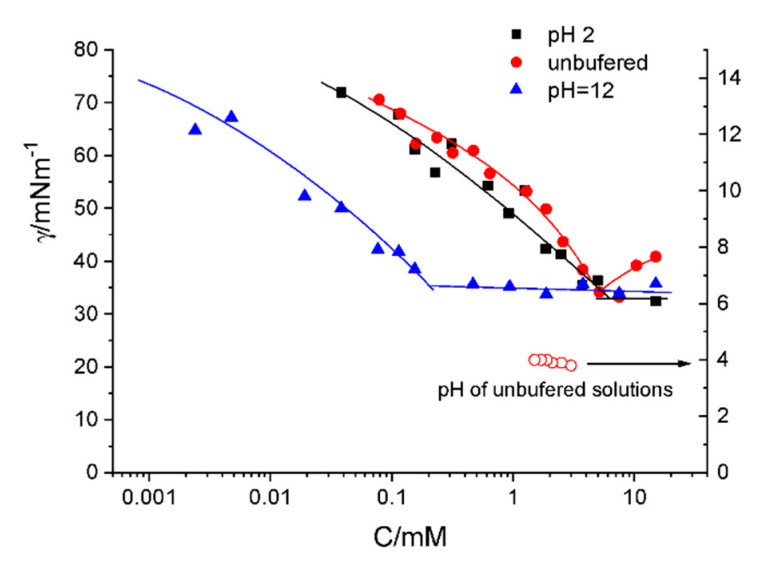
Surface tension as a function of concentration for histidine surfactant (DMHNHC_14_) in three different pH conditions at 25 °C.

**Figure 3 ijms-21-08912-f003:**
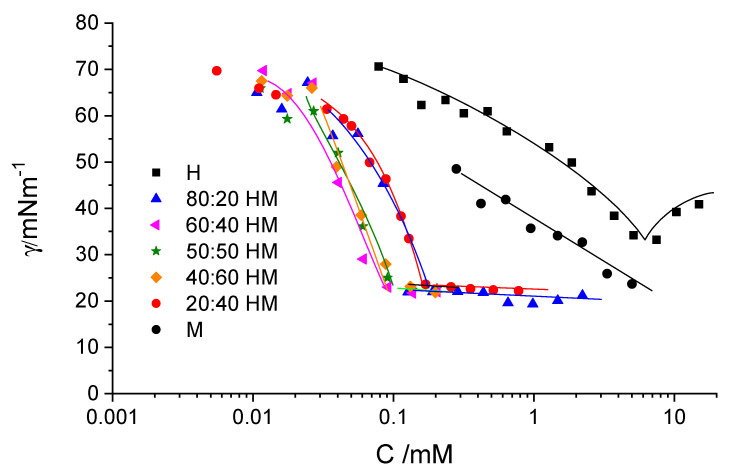
Surface tension as a function of total surfactant concentration for the different catanionic mixtures of DMHNHC_14_ and sodium myristate and the pure components measured at 25 °C.

**Figure 4 ijms-21-08912-f004:**
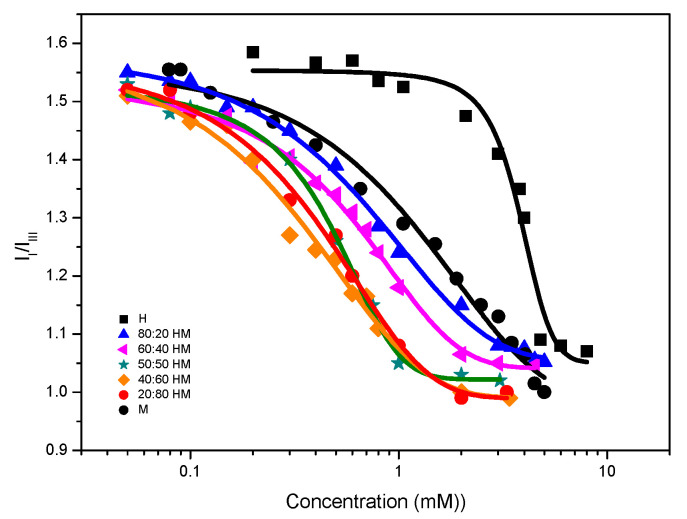
I_I_/I_III_ ratio obtained from pyrene fluorescence in the presence of the pure surfactants and the catanionic mixtures.

**Figure 5 ijms-21-08912-f005:**
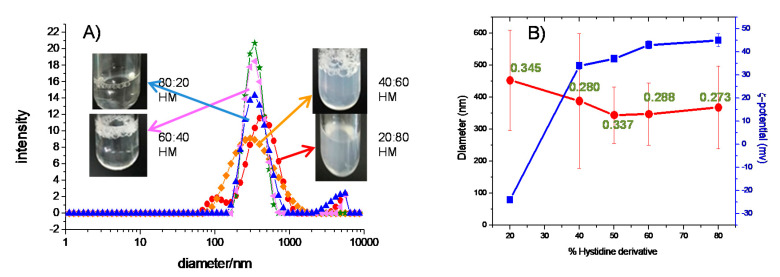
(**A**) Intensity weighed diameter obtained from dynamic light scattering (DLS) and visual appearance of the HM mixtures at 25 °C 
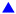
 80:20 HM, 
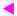
, 60:40 HM, 
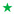
 50:50 HM, 
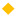
 40:60 HM, 
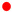
 20:80 HM. (**B**) mean DLS diameter (
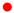
 left) and ζ-potential results (
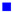
 right) for the HM formulations.

**Figure 6 ijms-21-08912-f006:**
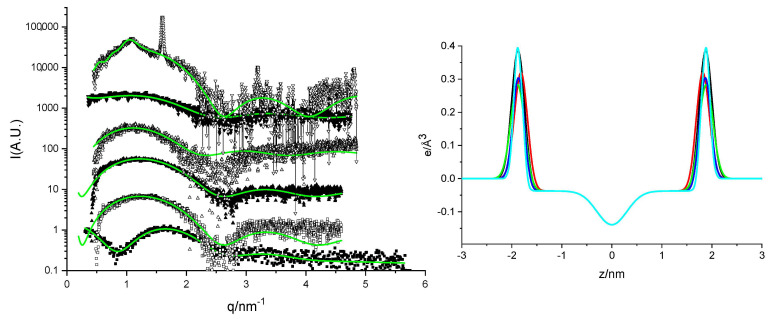
Left: SAXS spectra of DMHNHC_14_ (squares) and increasing contents of Myristate from bottom to top 80:20 HM, 60:40 HM, 50:50 HM, 40:60 HM and 20:80 HM. Fitting curves of lamellar models for the mixtures and core-shell spherical poly-disperse micelles for pure DMHNHC_14_ at 25 °C. The curves have been scaled by factors of five with the original intensity for the bottom one. Right: Electron density profiles of the mixtures as obtained from the fitting of the lamellar model on the SAXS spectra at 25 °C, lines in black 80:20 HM, red 60:40 HM, green 50:50 HM, blue 40:60 HM and light blue 20:80 HM. See the Methods Section for details of the model.

**Figure 7 ijms-21-08912-f007:**
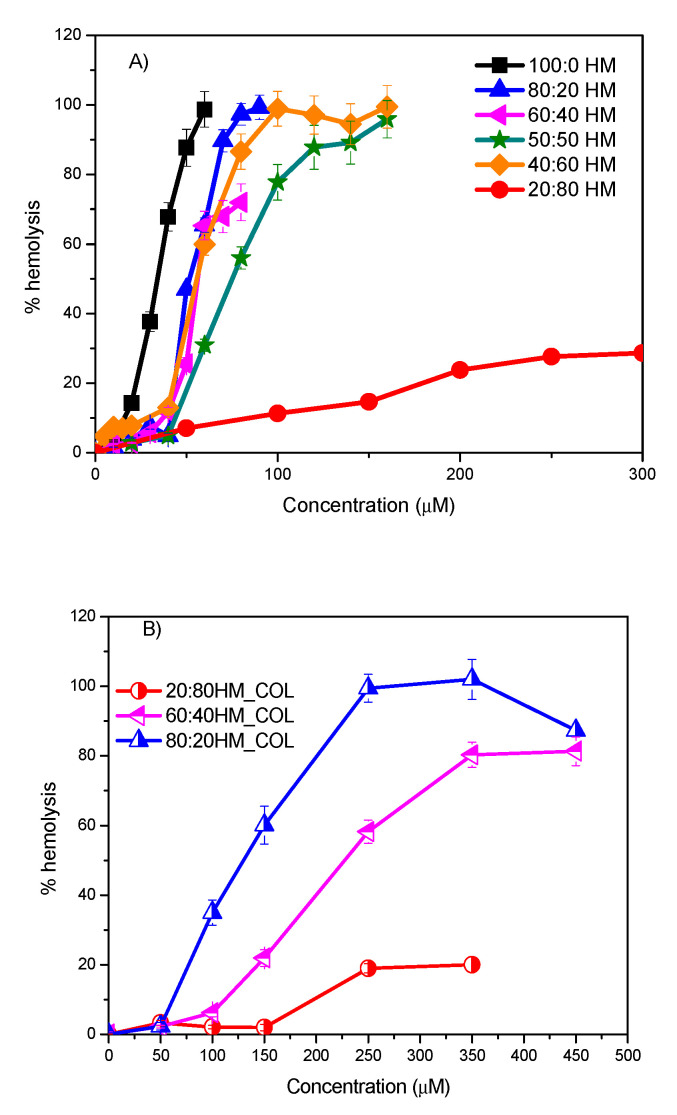
Plots of percentage of haemolysis versus concentration for (**A**) the pure DMHNHC14 and their HM mixtures and (**B**) the HM cholesterol containing formulations.

**Figure 8 ijms-21-08912-f008:**
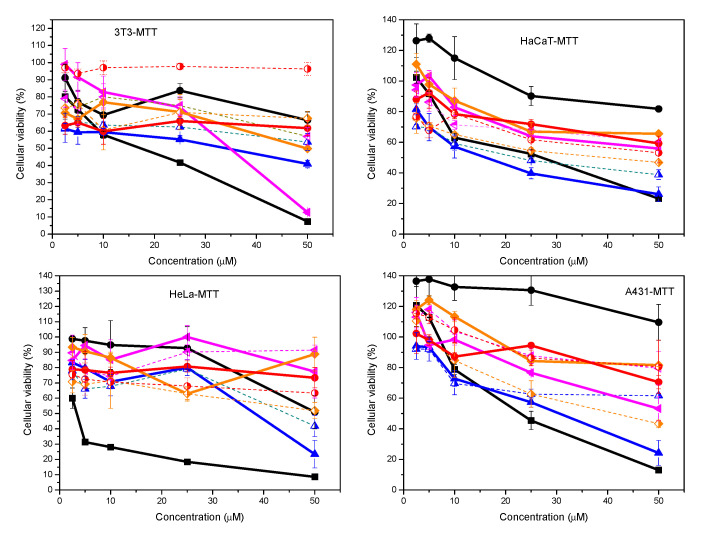
Cellular viability (%) as a function of concentration for different cellular lines using the MTT test: 
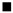
 Histidine, 
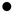
 Sodium myristate, 
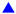
 80:20 HM, 
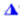
 80:20 HM_COL, 
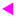
 60:40 HM, 
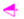
 60:40 HM_COL, 
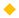
 40:60 HM, 
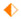
 40:60 HM_COL, 
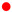
 20:80 HM, 
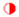
 20:80 HM_COL.

**Figure 9 ijms-21-08912-f009:**
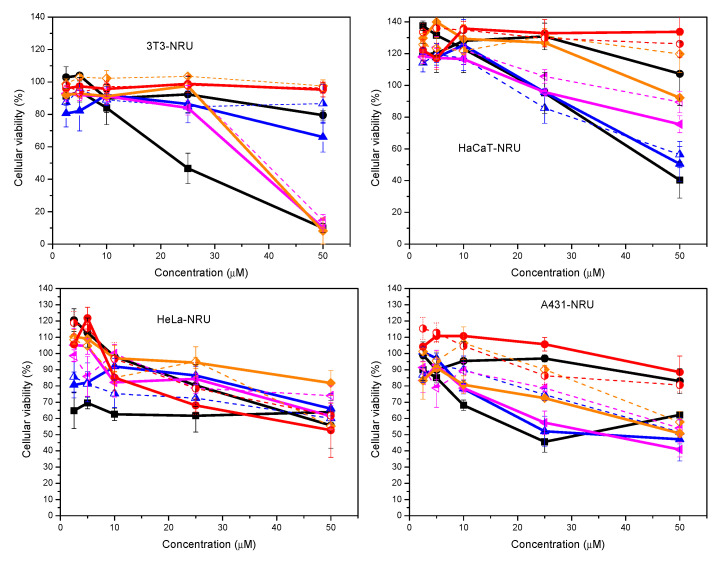
Cellular viability (%) as a function of concentration for different cellular lines using the NRU test: 
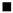
 Histidine, 
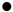
 Sodium myristate, 
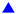
 80:20 HM, 
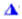
 80:20 HM_COL, 
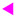
 60:40 HM, 
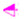
 60:40 HM_COL, 
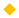
 40:60 HM, 
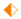
 40:60 HM_COL, 
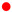
 20:80 HM, 
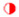
 20:80 HM_COL.

**Table 1 ijms-21-08912-t001:** Abbreviations and composition (%, molar) of the prepared catanionic mixtures.

	DMHNHC_14_ (%)	Sodium Myristate (%)		DMHNHC_14_ (%)	C_12_C_3_L (%)
80:20 HM	80	20	80:20 HL	80	20
60:40 HM	60	40	60:40 HL	60	40
50:50 HM 40:60 HM 20:80 HM	50 40 20	50 60 80	40:60 HL 20:80 HL	40 20	60 80
**Mixtures with Cholesterol**					
	Cholesterol (%)		DMHNHC_14_ (%)	Sodium Myristate (%)	
80:20 HM_COL	20		64	16	
60:40 HM_COL	20		48	32	
40:60 HM_COL	20		32	48	
20:80 HM_COL	20		16	64	

**Table 2 ijms-21-08912-t002:** Surface tension and fluorescence aggregation parameters of the pure surfactants and the catanionic mixtures and ζ-potential of the mixtures.

HM	A_m_ ^1^	CMC _γ_ ^2^	γ _CMC_ ^3^	α ^4^	β ^5^	CMC_F_ ^2^	α ^4^	β ^5^	ζ-potential ^6^
100	0.57	5.2	33			3.8			
80:20 HM	0.28	0.17	23.5	0.49	−11.3	0.60	0.49	−5.5	+45
60:40 HM	0.26	0.10	23	0.458	−12.5	0.47	0.43	−5.4	+43
50:50 HM	0.17	0.096	23	0.44	−12.3	0.29	0.41	−7.3	+36
40:60 HM	0.22	0.090	22	0.43	−12.6	0.31	0.39	−6.7	+33
20:80 HM	0.20	0.12	22	0.39	−11.9	0.40	0.32	−5.6	−24
Myr	1.0	5.0 ^7^	24			0.90			

^1^ Area per molecule (**A_m_**) according to Gibbs adsorption isotherm using *n* = 1 in nm^2^, estimated error is 15%. ^2^ CMC obtained by surface tension (**CMC****_γ_**) or Fluorescence (**CMC_F_**) in mM, estimated error is 10%. ^3^ Surface tension (**γ**) at CMC in mNm^−1^, estimated error is 1 mNm^−1^. ^4^ α DMHNHC_14_ molar fraction in the micelles. ^5^ Interaction parameter (β). ^6^ ζ-potential in mV, estimated error is 2 mV. ^7^ This is close to the solubility limit in our experiments.

**Table 3 ijms-21-08912-t003:** Minimum inhibitory concentration (MIC) of the pure histidine derivative and its HM catanionic mixtures. Values in brackets correspond to the concentration of DMHNHC_14_ in the catanionic formulations at their MIC values (MIC_H_). The MIC values of sodium myristate were higher than 113 μM for all microorganisms tested.

	MIC (μM)					
	DMHNHC_14_	80:20 HM	60:40 HM	50:50 HM	40:60 HM	20:80 HM
*Kocuria rhizophila* ATCC 9341	28	28 (22)	56 (34)	56 (28)	56 (22)	227 (45)
*Bacillus subtilis* ATCC 6633	28	28 (22)	56 (34)	56 (28)	113 (45)	454 (90)
*Staphylococcus epidermidis* ATCC 12228	14	56 (45)	56 (34)	28 (14)	113 (45)	227 (45)
*Staphylococcus aureus* ATCC 29213	28	28 (22)	28 (17)	28 (14)	56 (45)	227 (45)
*Klebsiella pneumoniae* ATCC 13883	113	227 (181)	227 (136)	> (>227)	> (>181)	> (>90)
*Escherichia coli* ATCC 25922	113	113 (90)	227 (136)	>454 (>227)	>454 (>181)	>454 (>90)
*Pseudomonas aeruginosa* ATCC 27853	227	>454 (>363)	>454 (>271)	>454 (>227)	>454 (>181)	>454 (>90)
*Candida albicans* ATCC 10231	28	28 (22)	56 (34)	56 (28)	56 (22)	113 (23)

**Table 4 ijms-21-08912-t004:** Minimum inhibitory concentration (MIC) of the pure histidine derivative and its HL catanionic mixtures. Values in brackets correspond to the concentration of DMHNHC_14_ in the catanionic formulations at their MIC values (MIC_H_). The MIC values of C_12_C_3_L were higher than 113 μM for all microorganisms tested.

	MIC (μM)				
	DMHNHC_14_	80:20 HL	60:40 HL	40:60 HL	20:80 HL
*Kocuria rhizophila* ATCC 9341	28	28 (22)	28 (17)	56 (22)	56 (11)
*Bacillus subtilis* ATCC 6633	28	113 (90)	113 (68)	113 (45)	113 (23)
*Staphylococcus epidermidis* ATCC 12228	14	113 (90)	113 (68)	227 (90)	454 (90)
*Staphylococcus aureus* ATCC 29213	28	113 (90)	113 (68)	113 (45)	227 (45)
*Klebsiella pneumoniae* ATCC 13883	113	227 (181)	454 (272)	>454 (>181)	>454 (>90)
*Escherichia coli* ATCC 25922	113	113 (90)	113 (68)	227 (90)	>454 (>90)
*Pseudomonas aeruginosa* ATCC 27853	227	>454 (>363)	>454 (>271)	>454 (>181)	>454 (>90)
*Candida albicans* ATCC 10231	28	56 (45)	113 (68)	113 (45)	113 (23)

**Table 5 ijms-21-08912-t005:** IC_50_ (µM) of the pure DMHNHC_14_, sodium myristate and their catanionic mixtures against the different cell lines studied in this work.

SYSTEM	Cellular Line							
	3T3		HaCaT		HeLa		A431	
	MTT	NRU	MTT	NRU	MTT	NRU	MTT	NRU
DMHNHC_14_	17.1	26.1	25.5	45.7	3.37	>50	22.6	>50
Myristate	>50	>50	>50	>50	>50	>50	>50	>50
80:20 HM	34.1	>50	15.5	>50	38.3	>50	29.5	26.9
80:20 HM_COL	>50	>50	21.8	>50	44.5	>50	>50	>50
60:40 HM	34.8	36.5	>50	>50	>50	>50	>50	36.1
60:40 HM_COL	>50	39.9	>50	>50	>50	>50	>50	>50
40:60 HM	50	38.3	>50	>50	>50	>50	>50	>50
40:60 HM_COL	>50	>50	>50	>50	>50	>50	>50	>50
20:80 HM	>50	>50	>50	>50	>50	>50	>50	>50
20:80 HM_COL	>50	>50	>50	>50	>50	>50	>50	>50

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
