# Peer review of "Aggregation Behavior, Antibacterial Activity and Biocompatibility of Catanionic Assemblies Based on Amino Acid-Derived Surfactants"

_ijms, 2020, doi:10.3390/ijms21238912_

Round 1

Reviewer 1 Report

This manuscript reports on the preparation of vesicles using cationic and anionic surfactants in an aqueous dispersion. This is an interesting concept. The manuscript is clearly written. The results are convincing, and the conclusions are supported by experimental data. The relevant literature is discussed. I believe that the manuscript can be accepted for publication after a minor correction concerning the following:

Why is there such a huge difference in zeta potential (table 2), if the molar fraction of DMHNH14 in micelles is similar?

If DMHNH14 is greater than what happened to the rest of the anionic surfactants?

Are catanionic vesicles actually formed when using C12H3L? what was the physical state of this anionic surfactant? Is it evenly distributed in the bilayer or does it form islands of anionic surfactant in the bilayer? How were these vesicles supported in solution? Is it possible to reproduce results from batch to batch with this surfactant?

Were these the same parameters of the vesicles when cholesterol was incorporated into the membrane? Was it the same bilayer thickness and vesicle sizes? DLS, zeta potential, and solution photos will help.

In addition, cholesterol makes membrane stiffer, how that can affect on parameters and properties of the vesicles, especially the charge distribution and interaction with/permeability of the membranes.

Reviewer 2 Report

The article is interesting, but requires refinement and systematization. Reading the works, one has the impression that it was written hastily. The sentences are long, they contain repetitions. The authors often cite their own works, which they do not write about directly. In my opinion it is incorrect.

  1. The work contains numerous writing errors. It is difficult to indicate the exact places because the lines are not numbered. Example: it says
    37  °C, it should be 37°C. Also, the % sign is stuck to a number once, other times it is written with a space.
  2. Point 2.2 page 13 Only one species of yeast was used in this study - Candida albicans. It is difficult to speak of antifungal properties on the basis of one strain. There is a lack of research on other species, including molds. Proposition - instead of antimicrobial activity, the title should be changed to antibacterial activity.
  3. Point 2.2 The same points as in the methodology should be considered; separately MIC, hemolysis, MTT and NRU tests.
  4. Page 15 The MIC value cannot be accurately determined by the serial dilution method in MHB medium if the test compounds (anionic compounds) are very slightly soluble in water.
  5. If the MIC values ​​were also determined for pure sodium myristate and C12C3L, they should be listed in Table 3 as > 454µM
  6. Why was the MIC value not determined for HM_COL compounds?
  7. Figure 10 The graph repeats the data shown in Table 3. It is not necessary. Additionally, there are no abbreviations and the name of abscissa axis.
  8. For which cells the IC50 was determined - no description in the methodology
  9. In the methodology, point 3.9. the bacterium Kocuria rhizophila is mentioned and Micrococcus luteus appears in the results
  10. The optimal growth temperature for Bacillus subtilis is 30°C, not 37°C, as stated by the authors
  11. Point 3.10 Hemolysis determination - What does mean concentrated solution of the cationic formulation? Specific values ​​should be provided.
  12. There is no explanation of abbreviations in the article, eg DMSO, FBS. All abbreviations used for the first time must be explained.
  13. Manufacturers of all reagents should be listed.
  14. Why are the concentrations of compounds given in mM in the physicochemical part, and in µM in the biological part? This is misleading.
  15. Graphical abstract is difficult to read, especially the chemical formulas. It is not known what the charts are about. Too much information.

Reviewer 3 Report

Manuscript ijms-981039

TITLE: Aggregation behavior, antimicrobial activity and biocompatibility of catanionic assemblies based on amino acid-derived surfactants.

AUTHORS: L. Pérez, A. Pinazo, M.C. Morán, R. Pons

The paper titled “Aggregation behavior, antimicrobial activity and biocompatibility of catanionic assemblies based on amino acid-derived surfactants” provides a description of mixed systems based on a histidine- and lysine-bearing surfactant mixtures with sodium myristate that involves a thorough aggregation characterization using tensiometry, fluorescent probe solubilization, an insight into the aggregate morphology using SAXS as well as biological activity study involving hemolysis, cell viability assays and antimicrobial activity assays. The work is of interest in connection with the use of natural amino acid surfactants, however, it requires major revision due to the following reasons:

  1. The manuscript mentions “to better control the self-assembly of these binary mixtures new functional surfactants are developed”. What specific goals should be achieved with surfactants to improve control the self-assembly?
  2. The phrase “CMC values clearly change when the surfactant positive charges go from 1 to 2 positive charges, however they are almost unnoticeable for three positive charges” requires a literary reference.
  3. Why is C12C3L considered anionic?
  4. It is necessary to indicate the composition of the buffer solutions in section 3, which were used for the study by tensiometry.
  5. How did the authors calculate the pKa value? I think it's better for the authors to present the distribution diagram of the protonated species of DMHNHC14 as a function of pH.

What about changes of spontaneous pH in mixed systems? The influence of pH on aggregation characteristics, e.g. cmc should be monitored for mixed systems. By the way, from the viewpoint of biological potentiality, physiological pH values would be of interest in this study.

  1. Please add details of methods especially how to prepare tested samples for tensiometry. What concentration of which surfactant was used to construct tensiometric dependences for mixed systems?
  2. Was there sediment or opalescence in mixed equimolar or other mixtures? If the precipitate formed after a certain period of time, please specify this time, and in the description of the experiments please indicate how long after the preparation of the solutions the measurement was carried out.
  3. Which component “seems not to form micelles directly at room temperature”?
  4. If “an open question is whether mixed micelles exist in the system or direct formation of vesicles from monomers are formed”, in order to close this issue, it is better to use TEM, rather than a mathematical model. The TEM images of the samples with pure surfactants also need to be displayed.
  5. Solvent used in NMR and chemical shift of reference must be defined. Resolution of Figure 6 needs to be optimized. On the basis of NMR spectra, the authors should indicate, between which chemical groups the interactions are realized for these surfactants.
  6. All figure readability and caption information should be improved.

For example, in figure 7, right picture, what do the yellow numbers mean?

Caption for figure 5 is too brief. Authors should correct it. Figure 5 shows the data for fluorescence of pyrene or of surfactant?

Figure 8 is not quite understandable. In this figure, is the Y-axis meaningful? Does the picture mean that HM produced a 10000 times more intensive scattering signal than DMHNHC14? If not, y-axis should be removed or formatted in a way to reflect the data.

The number of Figures in MS may be reduced and moved to SI, e.g. titration curve, etc.

  1. Authors are recommended to calculate the aggregation numbers by the pyrene quenching technique to compare with the aggregation numbers obtained by SAXS.
  2. How correct are the results obtained by SAXS for particles larger than 200 nm?
  3. How was the pure C12C3L composition prepared for biological research?
  4. What can be the reason for the “decrease of the cationic surface charge in the vesicles promoted by the addition of cholesterol”?
  5. The data in all tables are missing the error value.
  6. Since the authors are investigating the interaction of DMHNHC14 with two surfactants (sodium myristate and C12C3L), it is necessary to provide a comparative analysis of the aggregation behavior of sodium myristate and C12C3L in the mixture with DMHNHC14. How does the mixed aggregation of DMHNHC14–sodium myristate mixture differ from DMHNHC14– C12C3L? What surfactant is better to use in co-aggregation with DMHNHC14 for practical use?
  7. For data on antimicrobial activity, comparison with commercial preparation should be given. In Experimental section, bactericidal activity is mentioned, with no data given in Discussion section. In what form, monomer or aggregated surfactants demonstrate antimicrobial effect?
  8. English should be improved substantially. There are a considerable stylistic mistakes or awkward uses of English. What do the words "monocatenary" and “acid-soap particles” mean? It would greatly aid readability and clarity of the manuscript to have the revision by a native English speaker.
  9. Authors should take a close look at IJMS rules. The word "Figure" should begin with a capital letter, and in the list of references somewhere abbreviated, and somewhere the full names of the journals are given. Authors should check all figure captions in manuscript and SI.

In the light of the above, the manuscript is hard to read, needs to be revised and rewritten and it is not recommended for publication in the present form. However, the research topic is interesting, and I would recommend the authors to resubmit it after revisions. It would benefit the manuscript, if authors divided the discussion section into more subsections for easier navigation and perception of the data.

Round 2

Reviewer 3 Report

This paper is much better than its previous form, and it is probably publishable, but should be reviewed again in revised form before it is accepted. I have following continuing concerns about this work:

1.      The composition of the buffer solutions is also unknown. Lines 907-911 are missing in the pdf version of article.

2.      I’d like to see UV/vis spectra of DMHNHC14 in absence and presence of C12C3L and sodium myristate, would it be possible to add these to the supplementary information.

3.      The concentration of which substance is displayed on the X-axis in Figure 3?

4.      SAXS is used to study the structure of colloids in the 1 nm to 200 nm range (J Vis Exp. 2013; (71): 4160. doi: 10.3791/4160). How reliable is the information concerning the properties of the bilayer of particles which size larger than 200 nm?
